# A Quasi-Isotropic Probe for High-Power Microwave Field Measurement

**DOI:** 10.3390/s24186001

**Published:** 2024-09-16

**Authors:** Roman Kubacki, Dariusz Laskowski, Rafał Białek, Marek Kuchta

**Affiliations:** Faculty of Electronics, Military University of Technology, 00-809 Warsaw, Poland

**Keywords:** microwave measurement, sensors, small antenna, protection against microwave radiation

## Abstract

In the paper, a new design of a quasi-isotropic antenna for high-power electromagnetic (EM) field measurement is presented, along with its investigation into suitability. The measuring probe is intended for assessing pulsed microwaves, which cannot be measured by available meters due to the high value of electric field strength and short pulse duration. The measurement of such a strong field is required according to guidelines for protecting people against microwave fields, especially those emitted by radars. The proposed probe is based on the concept of dipole–diode detection. To enable high-power measurement, the receiving antenna is electrically “small,” allowing diode detection within the diode square-law characteristics range. Additionally, the shortened dipole length minimizes the spatial integration error, which can be significant in the case of microwave measurement. To obtain the desired antenna polarization, a new dipole geometry was proposed. Fulfilling the requirement of measuring all incident EM field polarizations, the receiving antenna was based on three dipoles arranged within a specific “magic” angular arrangement, ensuring a suitable quasi-isotropic radiation pattern. The proposed probe can operate in a frequency range from 1 GHz to 12 GHz.

## 1. Introduction

The development of contemporary civilization is associated with the greater use of various forms of electromagnetic energy. Considering that electromagnetic field (EMF) is not detectable by organoleptic methods (besides the optical frequency range), electromagnetic field detection requires the use of specific tools (meters). The emitted EMF is characterized by different parameters, among others: frequency, wave amplitude, polarization, and modulation. To measure the EMF field with appropriate accuracy, all these parameters must be considered. There is no universal measure; therefore, measurement apparatus should be adapted to determine the electromagnetic metrology requirement. To fulfill electromagnetic compatibility and health protection guidelines, the probe meter should be wide-band and isotropic, allowing detection of all rays incident from different directions. To complete the characterization of the electromagnetic field, the electric (E) as well as the magnetic (H) field strengths should be measured. However, when the measuring point is in the Fresnel or far-field regions, only E can be determined. The value of H, however, can be easily calculated from wave impedance (Z = E/H = 120π Ω). Of note, in the microwave frequency range, the transmitting antennas are electrical types, so a measurement of only E is sufficient. 

The dominat technique of E field measurement is the use of a probe equipped with an antenna (mainly a symmetrical dipole type) loaded by a detector (diode or, more rarely, thermocouple) with a lowpass filter and the subsequent transfer of DC voltage from the probe to an indicator (meter) through a transmission line. Nevertheless, there are some restrictions in designing the probe antenna. The initial problem with regard to the probe is due to the non-isotropic radiation character of a single dipole. To resolve this problem, three dipoles arranged into an equilateral triangle are typically used, hence creating a 3D probe. A second important probe antenna parameter is the length of the dipole. A long receiving antenna guarantees induced high electromagnetic force, but this is not used as an antenna probe for two reasons. The first is that long antennas are impractical for measuring E-field change in space—particularly due to the diverse antenna average values along the antenna length, as variable values of the field strength can be found in the vicinity of the antenna, i.e., in the near field, while a similar EMF variable character also takes place in the far field. Due to ray interferences, because of multi-path propagation, fast and slow fading exist. In this case, a long antenna probe causes amplitude and phase integration issues. 

The second reason for meter antenna minimization arises from the diode voltage requirement. Any induced voltage should be within the diode operation range. Low-induced voltages allow diode detection according to the square-law characteristic. Such diode “ableness” enables direct detection of power density, which is proportional to E^2^. For higher signals, the diode response is not square and needs to be digitally corrected. In the case of the Schottky diode that is typically used as a sensor, the square function is within a narrow voltage range, and in situations of high electric field strength, the diode can even be damaged. Indeed, the maximum measuring threshold for commercially available meters is 1 kV/m [1,2]. However, many civil and military devices emit higher pulse radiation, and the level of radiation produced as a pulse can be a thousand times higher than the root-mean-square (RMS) value. It must be underlined that pulsed radiation must be measured with regard to electromagnetic compatibility (EMC) as well as human protection regulations. Due to national and world guidelines, there exist limitations against high-power pulsed exposure [3,4,5]. The metrology and measuring thresholds of pulsed radar radiation per civil and military regulations can be found in [6].

In the paper, a new meter probe receiving antenna geometry is proposed and presented. This probe can be used for high-power EMF measurement. The dipole antenna was designed so as to fulfill the requirement of being “electrically small” over the entire dipole frequency range. The suggested quasi-isotropic antenna pattern was obtained via a three-dipole arranged within a certain angular pattern. Moreover, the probe transmission line is the resistive line—protecting against parasitic induction yet having a time constant low enough so as to enable the measurement of pulsed fields. The proposed solution of a short antenna with the developed geometry and function of current in the dipole has allowed the obtaining of flat electric probe properties in a frequency range from 1 to 12 GHz.

## 2. Electrically Small Receiving Antenna

A typical dipole–diode probe for E-field measurement is presented in Figure 1. Generally, such a probe consists of a receiving antenna, a diode, and a transmission line. 

When a dipole is used as a receiving antenna, the induced voltage (*e_A_*) depends on the incident electric field strength (*E*) and on the effective length of the antenna (*h_ef_*), according to the following relationship:(1)$$e_{A}=Eh_{ef}$$

The dipole effective length depends on measured field frequency, and, therefore, the probe needs an additional network so as to obtain flat frequency characteristics. Such frequency flat ability is realized by a transmission (feeding) line equipped with an RLC (resistor, inductor, and capacitor) network. However, the added RLC network brings about a higher probe time constant, making it impossible to measure pulsed modulated fields. From this point of view, most available meters cannot be used to measure the radiation emitted by radars. In the proposed solution, the resistive line used has a 500 Ω/cm value—allowing measurement of pulsed fields.

Another possibility for obtaining a flat-induced dipole voltage with regard to frequency function—and without incorporating an RLC network—is to shorten the dipole length. A short antenna, however, ensures lower voltage, but this scarcity can be easily filled by an added amplifier. 

The physical size of the meter-receiving antenna is an important limiting factor with respect to antenna performance. This limitation is usually manifested through electrical parameters such as efficiency, quality factor, bandwidth, etc. Better antenna performance can be obtained when an antenna has the resonant attributes that come with an overall dimension of half of a wavelength (λ/2). In this case, the antenna has good electric properties—but it is narrowband. 

The existence and development of different microwave measuring devices such as RFID (Radio-Frequency Identification), wearables, sensors, and meter probes demand the miniaturization of antennas. However, as an antenna is reduced in size, the antenna gain and efficiency will degrade. The answer to the raised question as to how small the antenna can be must take into consideration the real application as well as the operating frequency range. It is commonly accepted that if an antenna has overall dimensions of less than one-quarter of a wavelength (λ/4), it is often referred to as an electrically small antenna (ESA) [7,8,9,10,11,12]. 

Various different miniaturization techniques have been reported in the literature. A monopole antenna with a length of approximately 0.16 λ that is top-loaded and covers a frequency range from 1750 to 2500 MHz was proposed in [8]. In [13], a miniaturized antenna for global navigation satellite systems (GNSS) was put forward. This antenna, which is approximately 8 times shorter than the free-space wavelength, covers part of the lower L-band and enables the reception of various GNSS signals.

The first fundamental restraint on ESA size was introduced by Wheeler, who considered the limitation for the value of the Q factor [9]. Chu, in turn, analyzed the stored and radiated energy in an antenna and provided the following formula for deriving the Q factor of an ESA with *ka* size [14].
(2)$$Q=\frac{1}{ka}+\frac{1}{{\left(ka\right)}^{3}}$$
where

*k*—propagation constant, *k* = 2π/λ,*2a*—length of the dipole antenna.

Using the Q factor, the bandwidth of the antenna can be determined as follows:(3)$$\frac{∆f}{f}≅\frac{1}{Q}$$

In the case of a measuring probe, the amount of electric field integration imposed by the finite size of the receiving antenna must be determined. To consider this integration, the current distribution within the antenna should be compared with an antenna known to have a uniform current distribution. Such comparison yields two limiting factors—the amplitude and the phase errors. The phase error is most sensitive and can be defined as a function of *ka* [15]:(4)$$\delta_{ph}=0.5\left(1-\frac{2ka}{sin\left(2ka\right)}\right)$$

The phase integration error for ESA, determined according to (4), is presented in Figure 2.

Figure 2 reveals the level of phase error in a small antenna of a particular antenna length in relation to wavelength. Considering the threshold of 3% of phase error, the antenna length should not exceed *ka* = 0.3. For higher values of *ka*, the integration error quickly grows. In the microwave frequency range, the threshold *ka* = 0.3 yields a very short dipole, which guarantees several desirable probe features, among others, a frequency-flat probe response without resonance and with low integration error. However, such a structure significantly suffers from its electrical coupling with a transmission line. As a result, the current does not flow along the geometrical dipole direction, and this effect influences the polarization of the dipole. This effect is discussed and solved in the third chapter.

## 3. Quasi-Isotropic Radiation Pattern of Probe

A probe incorporating a dipole–diode, as presented in Figure 1, is easy to fabricate; however, such a probe element suffers from important electric disadvantages. The one-dipole antenna pattern is not isotropic and detects an electric field only when the E vector is parallel to the dipole. Such a directional aspect of measuring allows easy analytical correlation of electromagnetic field in the measuring point; however, this form of measuring is applied in a few meters due to important polarization drawbacks. Such a directional probe has a lower gain or even tends to zero when the received signal comes from no specific direction. To measure the incident field from an unknown direction when using a directional antenna, the probe alignment should be oriented to maximum display. In practice, to determine incident electric field strength, the three components should be determined in three orthogonal probe orientations, and the resultant value can be calculated via the following formula: (5)$$E_{res}=\sqrt{E_{p1}^{2}+E_{p2}^{2}+E_{p3}^{2}}$$
where

*E_res_*—the resultant value,*E_pn_*—component measured in n-th orthogonal probe orientation.

The measurement technique based on relation (5) needs three measurements at three different moments in time. Such an attempt is difficult to realize in practice because the amplitude, phase, and field configuration are strongly changeable in time, and the field at any moment can be different when compared to the next moment. Proper measurement, however, can be performed with a probe, allowing measurement of three electric field vectors at the same time. Thus, an effective probe should be isotropic. However, in current literature, divergences exist in the nomenclature definition of isotropic and omnidirectional. An isotropic antenna is a theoretical antenna that radiates/receives equal power in all directions and has a spherical radiation pattern. On the other hand, an omnidirectional antenna radiates power equally all around the antenna in any (e.g., horizontal) plane but only over a limited range of directions in the orthogonal (e.g., vertical) plane [16,17,18]. In this work, the term “quasi-isotropic pattern” is used because the metallic handle influences the spherical pattern. In effect, the pattern is not perfectly spherical but is spherical with an error of 3 dB.

The easiest way to build a suitable isotropic probe is to arrange three dipoles to be situated orthogonally to each other. In such a three-axis antenna, the three E-field components (e.g., *E_x_*, *E_y_*, *E_z_*) can be determined simultaneously. The dipole’s arrangement along the Cartesian axis is simple; however, there is a problem with the transmission line—which would be parallel to one of the dipoles and would operate (as being parallel to one dipole) as an additional receiving antenna. 

Another popular solution allowing the obtaining of an effective isotropic probe is to arrange three dipoles in the form of an equilateral triangle, with the angle being called a “magic angle.” The geometry of the magic angle and the schematic structure of the resulting omnidirectional antenna system are presented in Figure 3. 

The magic angle can be derived as the angle of the diagonal of the virtual cube and is representative of its three edges (L): (6)$$\theta =arcsin\left(\frac{L\sqrt{2}}{L\sqrt{3}}\right)\approx 54.74{}^{\circ}$$

## 4. The Polarization Problem with Small Antennas

Based on the presented conditions, one can expect that a probe of shortened length and an isotropic pattern can allow the detection of a strong microwave field in multi-path propagation conditions. However, as discussed, merely shortening the dipoles does not guarantee the desired linear polarization of the antenna due to the need to electrically couple the probe with the transmission line. Suitable isotropic radiation patterns can, however, be obtained when the current flows along the dipole in a manner that follows the magic angle. The problem is that despite the dipole geometry being arranged in a magic-angle form, the current can still flow in different directions. Figure 4 shows a simplified shortened dipole and that the current flow does not follow the magic angle despite the dipoles being designed to incorporate this angle.

A polarization type of antenna depends on the current function. This brings about a situation wherein the current within the simple shortened dipole is not oriented with the magic angle (Figure 4b); hence, an antenna system composed of three such dipoles cannot form an isotropic antenna pattern. To obtain the desired current direction in the dipole, there is a need to minimize the coupling effect with the transmission line. The proposed design of the dipole realizing this coupling minimization is presented in Figure 5a. 

In the modified dipole design, the arms are connected with the feeding line by a narrow wire placed at the upper part of the dipole part. In this configuration, the current flows along the magic angle on both arms—as demonstrated in Figure 5b. Such a dipole shape with current flowing along the magic angle was adopted to construct the isotropic antenna system.

## 5. Results

### 5.1. Probe Incorporating the Modified Antenna System

Utilizing the modified antenna shape, we fabricated an antenna system arrangement of three dipoles in an equilateral triangle configuration. The probe consists of three segments (Figure 6b). The single segment, presented in Figure 6a, is composed of a detection part (dipole and diode) and a resistive transmission line covered by a protective coat and ceramics.

Alumina ceramics type AL2O3 (Al_2_O_3_) are used in the proposed probe solution. The dipole size is a = 5 mm—fulfilling the requirement of being a small antenna functioning in the microwave frequency range. The transmission line is composed of resistive paste (DuPont-2009) and was coated to protect against atmospheric influence. The resistance of the resistive paste is 500 Ω/cm, and this value ensures that the induced current within the transmitting line due to strong EMF can be negligible and is easy to compensate for due to similar phases in each resistive path. With the electrical size of the dipole of *ka* = 0.3, such an antenna operates far from resonances, and there is no need to add any additional RLC correction network to obtain the flat transmittance. In the proposed solution, the added resistive line increases the value of the time constant; however, its value is still small enough to allow measurement of the pulsed radiation emitted by radars. 

### 5.2. Radiation Pattern

In the case of multi-path radiation, isotropic measurement is an important task. However, the measurement of all electric components must be realized simultaneously. In the proposed probe, the output signal is measured by each channel and multiplied by a suitable coefficient obtained through calibration (after calibration, each signal was found to be adequate to electric field strength). Finally, the resultant value is calculated as the Root Square Sum (RSS) of three values from all three segments.
(7)$$E_{res}=\sqrt{E_{s1}^{2}+E_{s2}^{2}+E_{s3}^{2}}$$
where

*E_sn_*—the value of *E* measured in the n-th probe segment.

To verify the isotropic characteristics of the probe, the radiation pattern (RP) should be determined. Considering the probe symmetry, the RP should be realized as two functions, i.e., in the functions of Theta and Phi, as shown in Figure 7.

In the verification of probe isotropic ability, both computer simulation and measurement were applied. Simulation was performed using CST Studio, and measurement was executed in an anechoic chamber (Figure 8). The probe was installed in a turntable with the vertical rotation axis passing through the dipoles. This guarantees the same phase of the incident field to the antenna system for all turning angles. 

Measurement of the antenna pattern was realized in the anechoic chamber by utilizing a turnover step of 15 degrees. At each measurement point, the signals from three segments were measured. The resulting RSS value was calculated according to (7). In Figure 9 and Figure 10, the antenna patterns are depicted at two frequencies, e.g., 2.45 GHz (Figure 9) and 10 GHz (Figure 10). The presented characteristics show the Theta and Phi functions of the resulting patterns. To make comparison between simulation and measurement easier, the patterns are shown as separate diagrams. 

To verify the isotropic property of the probe, simulations and measurements were conducted. These showed satisfying agreement. The obtained result confirms a quasi-isotropic pattern, the error being 3 dB. In measurements in the function of Theta for angles from 160 to 200 degrees, the probe was found to detect higher values due to the presence of the metal stand. Herein, the values are outside the diagram. Of note, in reality the meter probe cannot be used to detect rays coming from these angles because of the presence of the body of the operator, so this is not a problem. 

## 6. Conclusions

In the work, we present a proposed design of an antenna system for the construction of a superior electromagnetic measuring probe, as well as the investigation of its merits. The probe is specialized for assessing high-power pulse-modulated microwave radiation. Such measurements are required according to guidelines for people's protection against strong microwave fields, especially those emitted by radars. The currently available meters cannot be used for the measurement of such high electric field strengths nor of such short pulse duration radiation. The proposed probe is based on a dipole–diode detector. To enable high-power measurement, the receiving antenna is electrically “small,” allowing diode detection within the diode square-law characteristics range. Additionally, the adopted shortened dipole length minimizes the spatial integration error, which can be significant in microwave measurement. To obtain the desired antenna polarization, a new dipole geometry was put forward. By reducing dipole coupling with the feeding line, a desirable polarization as demonstrated by dipole current flow was obtained.

In environmental EMF measurement, an important element is the antenna being omnidirectional. Considering the multi-path propagation of EMFs, any isotropic antenna should measure all polarizations and all incident rays. So as to achieve this, we developed a receiving antenna system based on three dipoles. With the proposed dipole geometry, a satisfactory omnidirectional antenna system pattern was obtained. The computer simulation of the antenna pattern shows good agreement with measurements conducted in an anechoic chamber. 

## Figures and Tables

**Figure 1 sensors-24-06001-f001:**
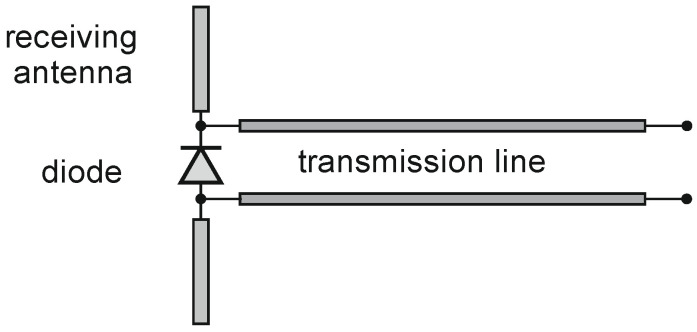
The dipole–diode schematic structure of an E-field probe.

**Figure 2 sensors-24-06001-f002:**
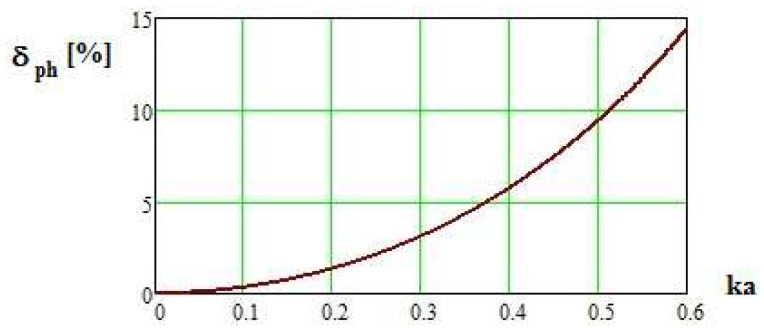
Phase integration error as a function of *ka*.

**Figure 3 sensors-24-06001-f003:**
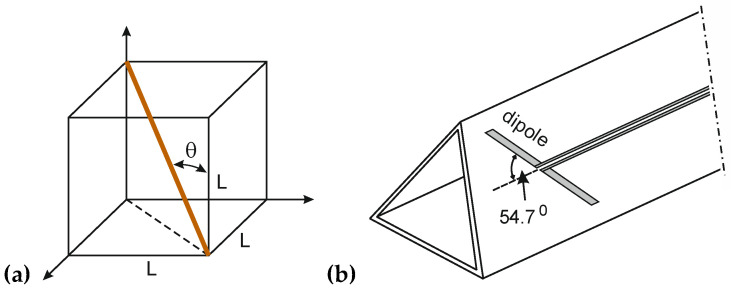
(**a**) Magic angle geometry; (**b**) diagram of antenna system.

**Figure 4 sensors-24-06001-f004:**
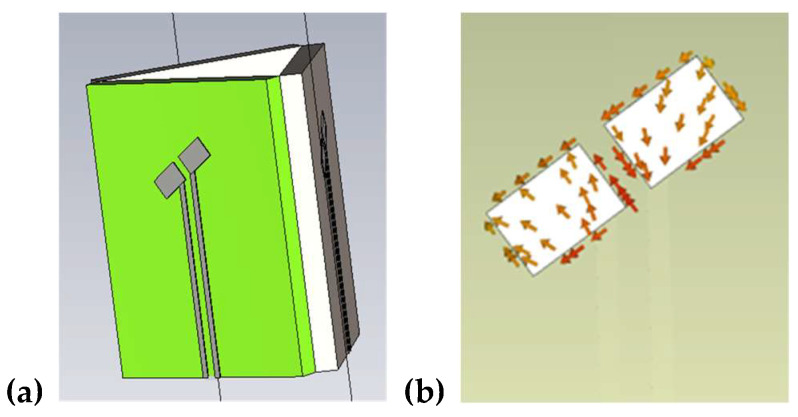
(**a**) Antenna system with small dipoles; (**b**) current flow within the dipole.

**Figure 5 sensors-24-06001-f005:**
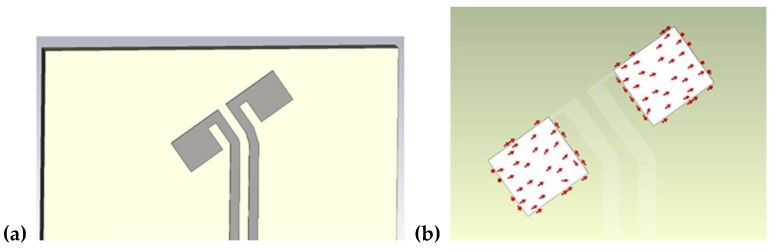
(**a**) Proposed shape of the dipole with transmission line; (**b**) current flow in the modified dipole.

**Figure 6 sensors-24-06001-f006:**
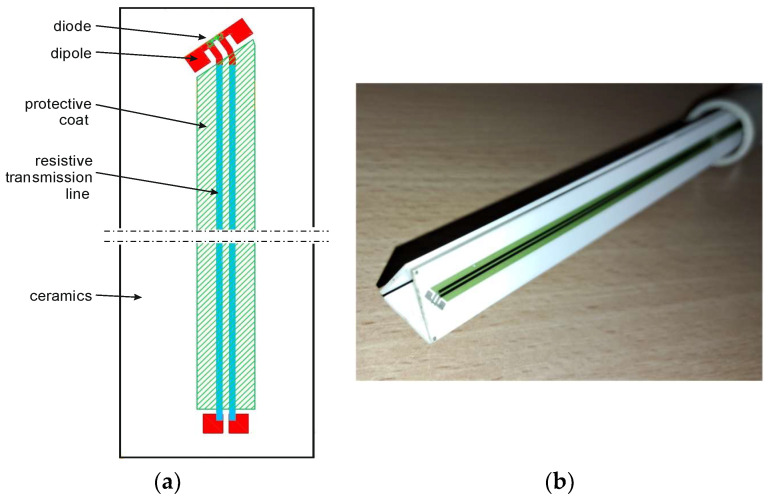
(**a**) Technical diagram of one segment of the probe; (**b**) picture of the fabricated probe.

**Figure 7 sensors-24-06001-f007:**
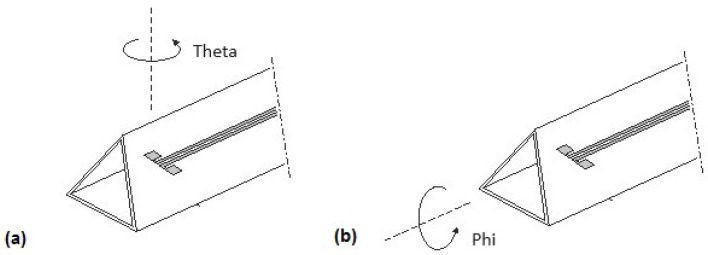
Probe rotations to determine the radiation pattern; (**a**) in function of Theta; (**b**) in function of Phi.

**Figure 8 sensors-24-06001-f008:**
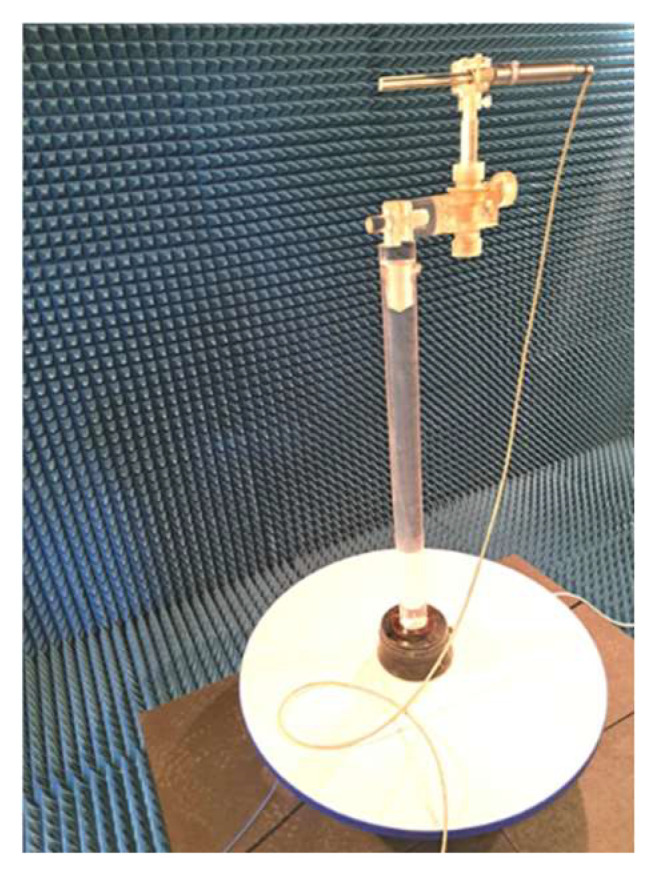
Probe measurement in an anechoic chamber.

**Figure 9 sensors-24-06001-f009:**
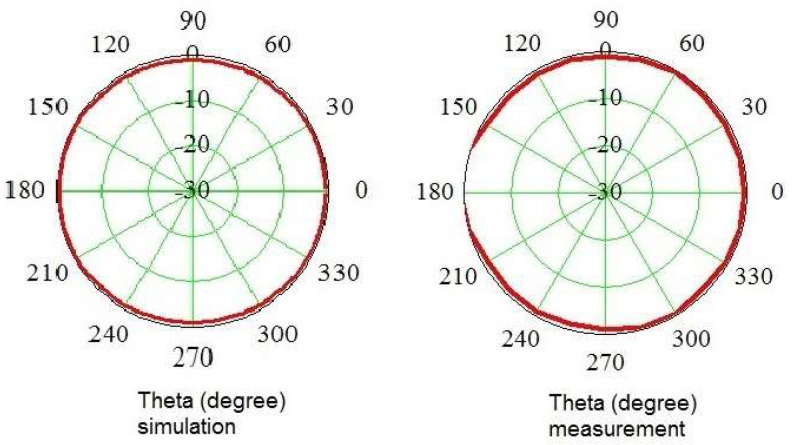
Simulated and measured antenna pattern at 2.45 GHz as a function of Theta and Phi. The left column shows simulated data, while the right reveals the measurement values.

**Figure 10 sensors-24-06001-f010:**
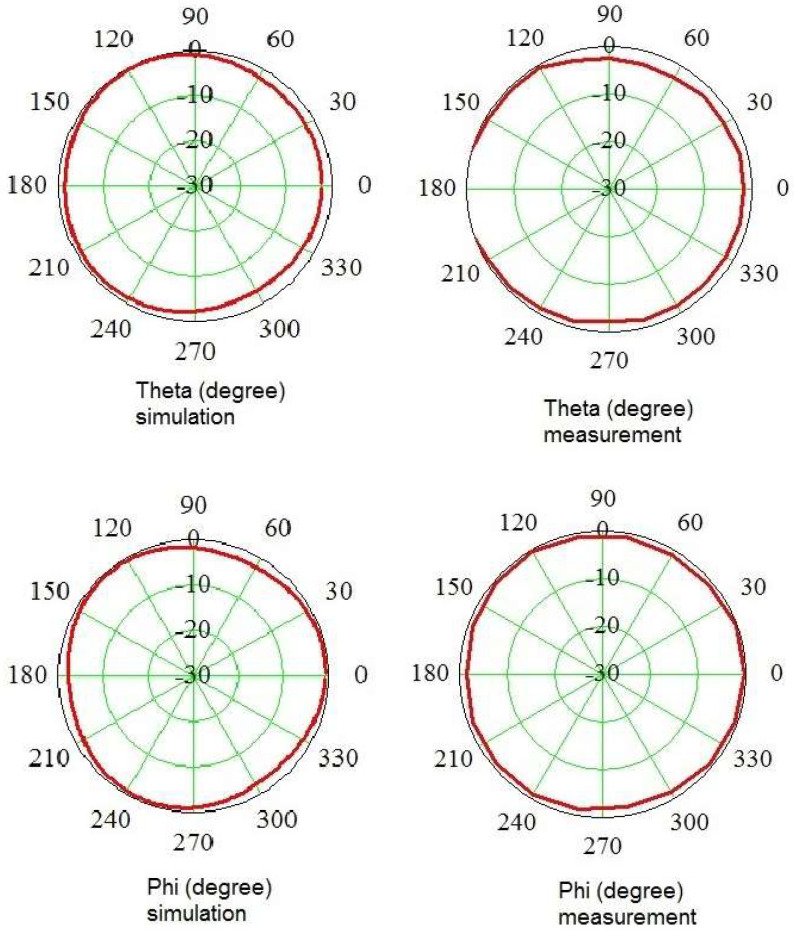
Antenna pattern at 10 GHz.

## Data Availability

The original contributions presented in the study are included in the article, further inquiries can be directed to the corresponding author.

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
