# Peer review of "A Quasi-Isotropic Probe for High-Power Microwave Field Measurement"

_sensors, 2024, doi:10.3390/s24186001_

Round 1
Reviewer 1 Report
Comments and Suggestions for Authors
In this manuscript, the authors presented the design of an antenna system for the construction of an EM measurement probe and investigated its merits. With the design of the dipole geometry, a satisfactory omni directional antenna system pattern was obtained. In addition, the simulation results are more consistent with the measured results. I recommend the manuscript for publication after addressing the following issues.
1. The scientific issues for the current development of microwave field measurement probes should be reviewed in the introduction part.
2. It is recommended that tables be added to compare the measurable electric field strengths of this work with other published works.
3. More relative publications are suggested to be cited to support views in this work.
4. What is the frequency range and detection bandwidth of the antenna?
Comments on the Quality of English LanguageMinor editing of English language required.
Reviewer 2 Report
Comments and Suggestions for Authors
List of references in Introduction should be appended with more basic paper on electric probes.
The language of the paper should be checked and corrected due to many errors and maybe words used incorrectly like:
"A long receiving antenna guarantees induced high electromagnetic force,"
Tt seems that better word for force would be signal.
The paper should also include information on the frequency range the probe can operate.
The comparison of the realized probe with any published data would increase the quality of the paper.
Comments on the Quality of English LanguageThe English language is of low quality. There are numerous simple grammar and style errors E.G. "three dipoles arranged into an equilateral triangle is typically used" a sentence from introduction should end with "are typically used".
Reviewer 3 Report
Comments and Suggestions for Authors
The manuscript presents a new design of an electromagnetic measurement probe based on a dipole-diode detector designed for measuring high-power pulse-modulated microwave radiation. The manuscript is well written and demonstrates fairly good agreement between the results of numerical simulations and direct experimental measurements for the proposed detector. The problem under consideration is of considerable practical interest, so I think that the manuscript in general deserves to be considered for publication in Sensors
